# Changes in Health-Related Quality of Life following Surgery in Patients with High-Grade Extremity Soft-Tissue Sarcoma: A Prospective Longitudinal Study

**DOI:** 10.3390/cancers16030547

**Published:** 2024-01-26

**Authors:** Anouk A. Kruiswijk, Michiel A. J. van de Sande, Cornelis Verhoef, Yvonne M. Schrage, Rick L. Haas, Marc H. A. Bemelmans, Robert J. van Ginkel, Johannes J. Bonenkamp, Arjen J. Witkamp, M. Elske van den Akker-van Marle, Perla J. Marang-van de Mheen, Leti van Bodegom-Vos

**Affiliations:** 1Department of Biomedical Data Sciences, Medical Decision Making, Leiden University Medical Center, 2333 ZA Leiden, The Netherlandsl.van_bodegom-vos@lumc.nl (L.v.B.-V.); 2Orthopedic Surgery, Leiden University Medical Center, 2333 ZA Leiden, The Netherlands; 3Department of Surgical Oncology and Gastrointestinal Surgery, Erasmus MC Cancer Institute, Erasmus Medical Center, 3000 CA Rotterdam, The Netherlands; c.verhoef@erasmusmc.nl; 4Department of Surgical Oncology, The Netherlands Cancer Institute, 1066 CX Amsterdam, The Netherlands; y.schrage@nki.nl; 5Department of Radiotherapy, The Netherlands Cancer Institute, 1066 CX Amsterdam, The Netherlands; r.haas@nki.nl; 6Department of Radiotherapy, Leiden University Medical Center, 2333 ZA Leiden, The Netherlands; 7Department of Surgical Oncology, Maastricht University Medical Center, 6229 ER Maastricht, The Netherlands; 8Department of Surgical Oncology, University Medical Center Groningen, 9713 GZ Groningen, The Netherlands; 9Department of Surgery, Radboud University Medical Center, 6525 EP Nijmegen, The Netherlands; han.bonenkamp@radboudumc.nl; 10Department of Surgical Oncology, University Medical Center Utrecht, 3584 CX Utrecht, The Netherlands; 11Safety & Security Science and Centre for Safety in Healthcare, Delft University of Technology, 2826 CN Delft, The Netherlands; p.j.marang-vandemheen@tudelft.nl

**Keywords:** soft-tissue sarcoma, extremity, health-related quality of life, physical functioning, mental health

## Abstract

**Simple Summary:**

This study investigates health-related quality of life changes in patients with a soft-tissue sarcoma during the diagnostic and treatment trajectory, and the differences in health-related quality of life changes between adults and the elderly since they face different challenges due to different levels of physical, social or work-related activities. Examining data from the VALUE-PERSARC trial, 97 patients completed the HRQoL questionnaires at diagnosis, and 3, 6 and 12 months thereafter. Results show comparable patterns across all measures, i.e., lower baseline scores, and a decrease at 3 months followed by subsequent improvement, reaching similar levels as the general population at 12 months. However, patients seem to struggle with the mental aspect of well-being, independent of age. The results of this study suggest that it is important to address both physical and mental health in the care of patients with a soft-tissue sarcoma.

**Abstract:**

Introduction: Changes in health-related quality of life (HRQoL) during the diagnostic and treatment trajectory of high-grade extremity soft-tissue sarcoma (eSTS) has rarely been investigated for adults (18–65 y) and the elderly (aged ≥65 y), despite a potential variation in challenges from diverse levels of physical, social, or work-related activities. This study assesses HRQoL from time of diagnosis to one year thereafter among adults and the elderly with eSTS. Methods: HRQoL of participants from the VALUE-PERSARC trial (*n* = 97) was assessed at diagnosis and 3, 6 and 12 months thereafter, utilizing the PROMIS Global Health (GH), PROMIS Physical Function (PF) and EQ-5D-5L. Results: Over time, similar patterns were observed in all HRQoL measures, i.e., lower HRQoL scores than the Dutch population at baseline (PROMIS-PF:46.8, PROMIS GH-Mental:47.3, GH-Physical:46.2, EQ-5D-5L:0.76, EQ-VAS:72.6), a decrease at 3 months, followed by an upward trend to reach similar scores as the general population at 12 months (PROMIS-PF:49.9, PROMIS GH-Physical:50.1, EQ-5D-5L:0.84, EQ-VAS:81.5), except for the PROMIS GH-Mental (47.5), where scores remained lower than the general population mean (T = 50). Except for the PROMIS-PF, no age-related differences were observed. Conclusions: On average, eSTS patients recover well physically from surgery, yet the mental component demonstrates no progression, irrespective of age. These results underscore the importance of comprehensive care addressing both physical and mental health.

## 1. Introduction

Sarcomas are a rare group of malignant tumors arising from connective tissue accounting for approximately 1% of adult malignancies [1]. With more than 70 histological subtypes, sarcomas are a diverse group of tumors that affect people of all ages and can occur at any anatomical site [2]. The estimated incidence in Europe per year is around 5 per 100,000 persons [3]. In the Netherlands, about 480 new patients per year are diagnosed with a high-grade soft-tissue sarcoma (STS) of which the majority occur in the extremities (42%), with an overall survival of 50% at 5 years [4].

The treatment of sarcoma patients primarily aims to provide uncomplicated local control in the extremity and to improve overall survival, typically achieved through surgery, often combined with (neo)adjuvant radiotherapy. In addition to prolonged survival, cancer patients prioritize the enhancement of health-related quality of life (HRQoL) as a critical factor in tumor treatment [5,6]. HRQoL is a multidimensional concept that includes the patient’s perspective on the physical, mental, social and cognitive domains of well-being [7]. HRQoL has emerged as an essential component of patient-centered care and outcome evaluation, providing valuable insights into the impact of sarcoma and its treatments on diverse aspects of patients’ lives.

Currently, the available literature on the HRQoL among sarcoma patients is limited, but has been expanding over recent years. However, most studies are cross-sectional in nature, offering only a momentary snapshot into the well-being of patients at a specific point in time [8,9,10]. Only the studies of Parades (2011) and more recently Eichler (2023) provide some insight in the longitudinal HRQoL, but both included a widely heterogeneous population, consisting of gastrointestinal stromal tumors and bone and soft-tissue sarcomas at various locations [11,12]. Furthermore, these studies do not give insight in the differences between patients of different age groups, even though the challenges faced by cancer, and particularly those associated with the treatment of the tumor, vary significantly depending on patient’s age. For instance, adults (18–64 years) often encounter heightened emotional, cognitive and social difficulties at the time of diagnosis, during and after treatment when compared to their elderly counterparts. This is underscored by findings indicating that adults are more prone to report concerns regarding recurrence and side effects of treatment than elderly patients [13]. Moreover, the age disparity manifests in the realm of daily activities, with adults facing more difficulties in sustaining their routine tasks. This may be attributed to the fact that adults typically engage in higher levels of physical activity, along with increased involvement in work-related and social activities compared to the elderly [13,14,15]. Notably, work-related activities hold particular significance for adults, serving as a crucial aspect of their well-being even throughout the diagnostic and treatment phases. In contrast, the majority of individuals aged 65 and above are either retired or approaching retirement, indicating a distinctive shift in priorities during the same trajectory.

Therefore, the aim of this study is to evaluate changes in HRQoL from time of diagnosis to one year thereafter of a homogeneous high grade extremity sarcoma patient population, separately addressing adult (aged 18–65 years) and elderly (aged ≥65 years) patients. 

## 2. Methods

The data used in this study were gathered during the VALUE-PERSARC trial, which has been described previously [16]. In short, the VALUE-PERSARC trial is a multicenter parallel cluster randomized controlled trial (RCT) evaluating the impact of the use of a risk-prediction model (PERSARC) during clinical decision making on knowledge and decisional conflict among eSTS patients. Patients aged ≥18 years with a (histologically confirmed) high-grade (Fédération Nationale des Centres de Lutte Contre le Cancer [FNCLCC] grade II and III [17]) eSTS, who were treated with curative intent, were eligible for inclusion. Patients treated without curative intent or patients requiring other treatment modalities than surgery and/or radiotherapy were excluded.

The VALUE-PERSARC trial has been registered online (NL9160/NCT05741944) and approved by the Medical Ethical Committee Leiden-Den Haag-Delft (METC-LDD) (NL76563.058.21).

### 2.1. Patients and Data Collection

All patients included in the VALUE-PERSARC trial from 1 August 2021 until 1 December 2023 were eligible for HRQoL assessment. The first assessment was completed at time of diagnosis, one week after the treatment decision had been made (T1). The second assessment, at 3 months after the treatment decision, was completed shortly after surgery and/or (neo)adjuvant radiotherapy (T2). At 6 months after the treatment decision, patient completed the third assessment (T3). The last assessment was completed at 12 months after the treatment decision was made (T4).

### 2.2. Health-Related Quality of Life

The PROMIS Global Health (v1.2), PROMIS Physical Function (v1.2) and EQ-5D-5L were used to assess patients’ HRQoL status [18,19,20]. Both measures, i.e., PROMIS and EQ-5D-5L, are not disease-specific and therefore universally applicable across various populations [21,22]. The PROMIS Global Health (GH) and PROMIS Physical Function (PF) are both short forms, each consisting of 10 items on a 5-point scale (e.g., never = 1, always = 5). PROMIS GH provides two scores, one for Physical Health and one for Mental Health, while the PROMIS PF generates a single score specifically for physical functioning, all on a scale from 0 to 100 on the T-metric. The EQ-5D-5L consists of two scores, (1) a descriptive system comprising the dimensions mobility, self-care, usual activities, pain/discomfort and anxiety/depression on a 5-point scale (no problems (1)—extreme problems (5)), resulting in one index score, and (2) EQ-VAS: a visual analogue scale, ranging from 0 (worst possible health) to 100 (best possible health), indicating patients self-rated health [23]. References values are accessible for EQ-5D-5L (mean + SD: 0.869 ± 0.170), EQ-VAS (82) and PROMIS (mean + SD = 50 + 10), with the latter measures standardized to the general US population [19,23,24]. The PROMIS manual suggests using these US parameters due to a lack of cross-validation specific to the Netherlands [25]. A 3-point difference in T-score was considered clinically meaningful, as found in previous studies among cancer patients using PROMIS, so that a clinically relevant reduced HRQoL for PROMIS GH and PF was defined by scores lower than 47 [26,27]. For the EQ-5D-5L and EQ-VAS, clinically relevant differences of, respectively, 0.06–0.08 and 7 mm have been previously reported in lung cancer patients [28]. In this study, we use a value of 0.07 to indicate a clinically relevant difference, i.e., a score < 0.799 to indicate reduced HRQoL for the EQ-5D-5L.

### 2.3. Statistical Analyses

Baseline characteristics were described using frequencies and percentages for categorical variables and mean (standard deviation) or median (interquartile range) for continuous variables, depending on the distribution.

Average HRQoL (mean + SE) was computed for the EQ-5D-5L and both PROMIS measures at the different timepoints among all patients, and per age-group (patients aged 18–65 years and patients aged ≥65 years). In addition, average HRQoL scores for the EQ-5D-5L and PROMIS measures will be compared between eSTS patients (overall and per age group) and the general population mean to identify clinically relevant differences. 

All analyses were conducted using R software, version 4.1.3 (R Foundation for Statistical Computing, Vienna, Austria) [29].

## 3. Results

### 3.1. Patients Characteristics

Between August 2021 and December 2023, a total of 97 patients were included in the VALUE-PERSARC trial and were invited to complete the HRQoL questionnaires. Characteristics of patients who completed the HRQoL questionnaires are shown in Table 1. Median age at inclusion was 64 (IQR 50–72) and 50% were female. The majority of patients were diagnosed with a myxofibrosarcoma (27%), (myxoid) liposarcoma (23%) or malignant fibrous histiocytoma/undifferentiated pleomorphic sarcoma (MFH/UPS) or soft-tissue sarcoma not-otherwise-specified (NOS) (22%). Most tumors were deep-seated (67%) with a median tumor size of 9 cm (IQR 5–12). The lower extremities were the tumor site most affected (80%). A total of 82 patients were treated with neo-adjuvant radiotherapy (RT) (85%) followed by surgery with free resection margins (R0) (56%). In general, pre-operative RT started one week after the treatment decision was made, following the standard procedure of 25 × 2 Gy (total duration of 5 weeks). This was followed by a recovery period of approximately one month, so that the majority of patients underwent surgery 10–12 weeks after the initial treatment consultation. Thirteen patients (13%) had surgical complications requiring reoperation, mostly related to wound infection or wound healing problems. Eleven patients experienced distant metastasis. Ten patients were deceased or lost to follow-up. 

### 3.2. Overall HRQoL Mean Scores over Time

Mean PROMIS GH, PROMIS PF and EQ-5D-5L scores for the different timepoints are reported in Table 2. At baseline, eSTS patients on average had clinically relevant reduced scores for EQ-5D-5L and PROMIS measures (T-score < 47, EQ-5D-5L < 0.829). Over time, similar patterns are observed in all measures, i.e., a decrease at 3 months (T2), followed by an upward trend to reach similar scores as the general population at mean 12 months, except for the mental component of the PROMIS GH, where scores remain lower than the general population mean (Figure 1). 

### 3.3. HRQoL Mean Scores over Time Stratified by Age

The stratified analysis for patients aged <65 and ≥65 years shows a similar trend in PROMIS PF up to and including the first 6 months (T3) (Figure 2). However, at 12 months, the PROMIS PF scores of patients aged <65 years were above the general population mean (T = 53.0), while scores for patients aged ≥65 years remained worse than the general population mean (T = 47.4), although this difference is not clinically relevant. The physical component of the PROMIS GH, however, did not show this difference at 12 months for patients aged <65 and ≥65 years (<65 = 50.7, ≥65 = 49.4), but showed a similar trajectory across both age groups, i.e., clinically relevant reduced scores at baseline followed by a decrease at 3 months with an upward trend exceeding the baseline similar to the general population mean (Figure 3).

The mental component scores of the PROMIS GH for patients remain somewhat comparable and lower than the general population mean throughout the entire trajectory (Figure 3). Although a difference is evident at 6 months (T3) between patients <65 and ≥65 (49.0 vs. 46.1), this difference disappeared at 12 months (47.9 vs. 47.2), with both scores not indicating a clinically relevant difference with the general population mean. Similar trends over the trajectory of the disease are seen for the EQ-5D-5L as observed for the mental component of the PROMIS GH, with no clinically relevant age-related differences at 12 months. Interestingly, unlike all other measures, patients’ self-reported health (EQ-VAS) is higher at baseline for patients ≥ 65 compared to <65 years, but this difference is no longer seen over time 6 months after diagnosis. Patients < 65 report even better EQ-VAS scores at 12 months than patients ≥ 65 years. Mean PROMIS GH, PROMIS PF and EQ-5D-5L scores for the different timepoints for patients aged <65 and ≥65 years are reported in Table 3.

## 4. Discussion

Our results show that high-grade eSTS patients, on average, recover well physically from the local management of their primary tumors, as reflected in scores surpassing baseline values at 12 months. Conversely, the mental component exhibits minimal progress during the corresponding timeframe. Age-related differences are observed in the PROMIS PF, where patients < 65 continue to improve with scores at 12 months above baseline, better than the general population; patients ≥ 65 improved less rapidly, especially between 6 and 12 months. However, this age-related difference was not observed in the physical component of the PROMIS GH. Interestingly, no age-related differences were observed in the PROMIS GH, as both age groups displayed minimal improvement throughout the trajectory from diagnosis to one year thereafter. 

### 4.1. Results in Context

Prior studies have reported reduced physical functioning scores in STS patients, but they did not measure mental health per se. Additionally, most of these studies used a cross-sectional approach, often assessed within a large time period (0–5 years post treatment) [8,9,10,30], whereas our study follows the trajectory of the disease and describes the impact that diagnosis and treatment have on HRQoL at specific time points up to one year after diagnosis. 

The prospective studies of Parades (2011) and Eichler (2023) are, to our knowledge, the only studies that also assessed HRQoL in sarcoma patients during different phases of the disease. Although Eichler et al. were able to include and follow-up a large number of patients, their population included sarcomas of any entity followed over a wide time interval (0–5 years), making comparisons with our study results challenging. Furthermore, they reported EORTC-QLQ-C30 domain scores as factors and not as mean score, which further complicates the comparison of the results of Eichler’s study with those of our study. Parades et al. assessed HRQoL in bone and soft-tissue sarcoma patients (*n* = 36) at diagnosis and 4 months thereafter, during which the majority of most patients still received chemotherapy, known to have a profound impact on the HRQoL [8]. Nonetheless, similar reduced HRQoL scores were found in our study for the period up to and including 3 months after diagnosis, which includes the diagnostic and treatment phase in extremity STS patients included in our study. In addition to the study of Parades, our study shows that physical functioning restores scores to above baseline levels similar to the general population within the first year post diagnosis. This improvement can be attributed to healthcare providers predominantly concentrating on patients’ physical symptoms and physical health [31]. In addition to findings that extremity STS patients generally experience robust physical recovery following surgery, our study shows that the mental recovery lags behind, showing minimal improvement throughout the trajectory from diagnosis to one year thereafter. Although the mental health scores 12 months after diagnosis in both age groups were just not clinically relevantly reduced, the overall trend does show lower scores than the general population. This underscores the importance of prioritizing efforts to address mental health issues. Elderly sarcoma survivors reporting inferior physical recovery after treatment compared to adults aligns with findings from prior research conducted by Drabbe et al. (2021), who compared HRQoL between survivors and the general population according to age approximately five years after diagnosis. Unfortunately, this study did not assess mental health itself, as it used the EORTC-QLQ-C30 to examine HRQoL, making comparison impossible. Nevertheless, the observed reduced mental health scores in our study emphasize the importance of placing increased attention on addressing mental health issues. It is imperative to note that further research, employing more comprehensive instruments, is necessary to gain a deeper understanding of the mental health of STS patients.

### 4.2. Strength and Limitations

To our knowledge, this study is the first to measure HRQoL at multiple timepoints after treatment in in a specific group of high-grade eSTS patients, who generally undergo uniform treatment. This enables us to gain a specific understanding of the HRQoL of eSTS patients from diagnosis to one year thereafter. Furthermore, the inclusion of patients from various sarcoma specialist centers across the Netherlands, along with the use of several generic questionnaires to assess HRQoL all showing comparable trends, enhances the generalizability of our results. 

This study is limited by potential selection bias due to the inclusion of patients in the VALUE-PERSARC trial. To participate in this study, patients must be invited by their clinician to participate, understand Dutch and must be willing to download the VALUE-PERSARC app on their personal device, which particularly excludes patients who are not adept with technology. This selection could potentially impact our outcomes and might explain why we did not find differences in PROMIS GH between age groups, where we expect primarily older, healthy patients to participate, thereby potentially overestimating HRQoL. Second, due to the small sample size, we could not evaluate the impact of other treatment- (e.g., surgical complications) and tumor-related (e.g., depth, location) factors on HRQoL. Another limitation is that the study was not powered to detect differences in HRQoL changes, as this was a secondary outcome in the VALUE-PERSARC trial. Therefore, we focused on describing the general patterns. Additionally, the specific timepoints to assess HRQoL were not linked to the date of surgery but to the treatment consultation, potentially causing differences in the HRQoL scores (at 3, 6 and 12 months) due to differences in care trajectories of patients who did not undergo RT or who underwent post-operative RT. Finally, short forms of the PROMIS Global Health and PROMIS Physical Function were used to reduce questionnaire burden; however, these measures do not capture all of the dimensions of HRQoL (e.g., social functioning and anxiety), making comparisons with the literature challenging. 

### 4.3. Implications

Longitudinal HRQoL data are rare in sarcoma patients; our results provide first insights into the trajectory from diagnosis till one year thereafter of eSTS patients. Based on our results, more attention should be paid to the mental health of patients with high-grade eSTS after initial treatment. At that time, patients may shift their focus from “fighting cancer” to “cancer survivor”, may have fear of recurrence, and may perceive a loss of support from healthcare providers as well as friends and family, which all have an profound impact on mental health [32]. Neglecting mental health during this period could jeopardize the effectiveness of healthcare and thereby adversely affect the HRQoL of STS patients [33]. Hence, it is essential to prioritize screening for mental health issues alongside the monitoring of physical health, and efforts should be made to integrate mental health screening both during active cancer treatment and survivorship.

## 5. Conclusions

Patients with soft-tissue sarcoma in their extremities show robust physical recovery after surgery; however, their mental health scores lag behind, with no age-related difference observed. Overall, healthcare providers should pay greater attention to the mental health of patients after initial treatment, as mental health influences physical health and overall quality of life, and vice versa. Addressing mental health can contribute to a more comprehensive and effective approach to their HRQoL during the diagnosis and treatment of soft-tissue sarcoma patients.

## Figures and Tables

**Figure 1 cancers-16-00547-f001:**
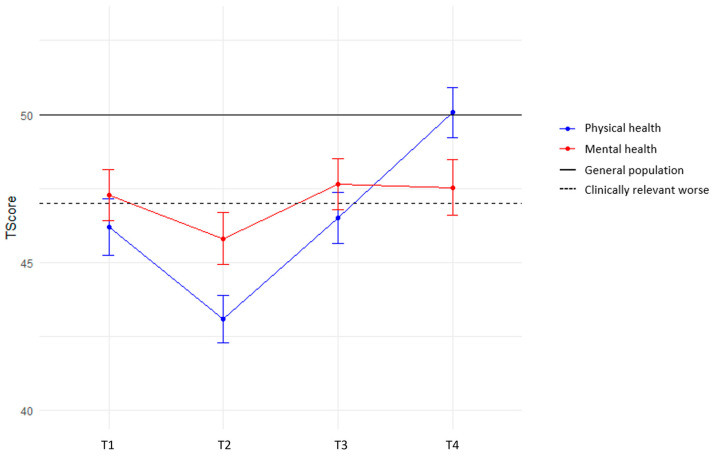
PROMIS Global Health mean scores (+SE) over time. The 4 timepoints are represented on the x-axis. Mean PROMIS T-scores are represented on the y-axis. T = 50; general population mean. T = 47; clinically relevant worse than general population. Blue = physical health, red = mental health.

**Figure 2 cancers-16-00547-f002:**
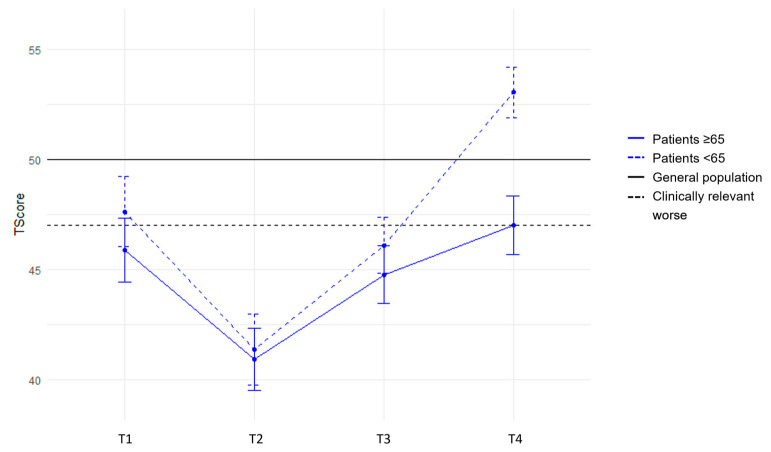
PROMIS Physical Functioning mean scores (+SE) over time stratified for patients aged <65 and ≥65 years. The 4 timepoints are represented on the x-axis. Mean PROMIS T-scores are represented on the y-axis.

**Figure 3 cancers-16-00547-f003:**
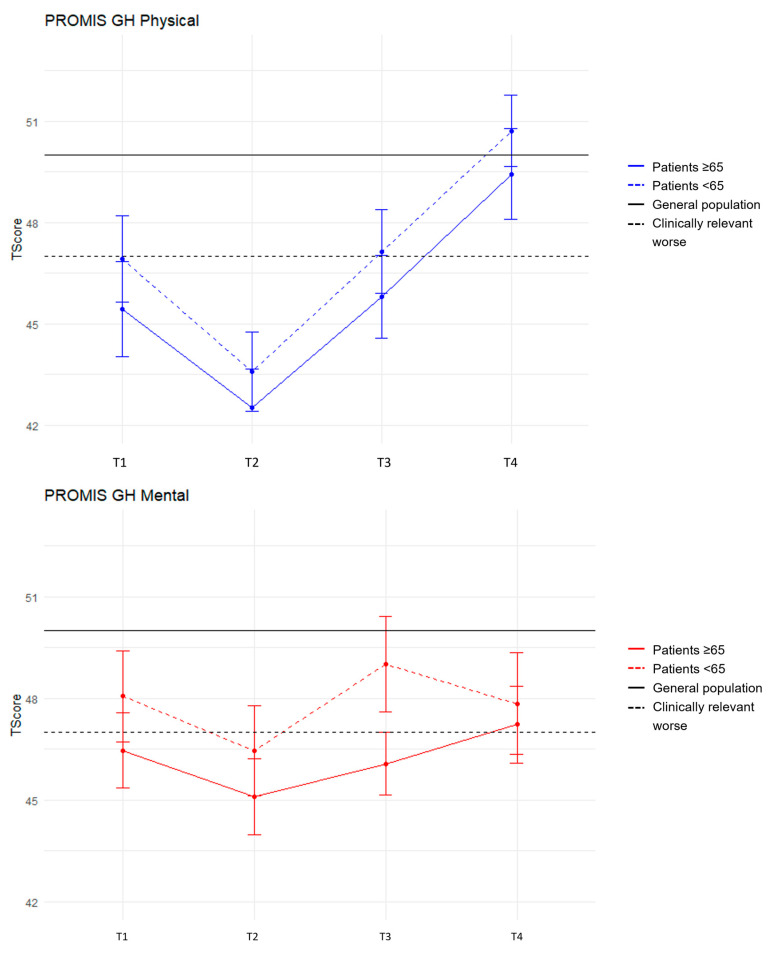
PROMIS Global Health mean scores (+SE) over time stratified for patients aged <65 and ≥65 years for the physical (**above**) and mental (**below**) domains. The 4 timepoints are represented on the x-axis. Mean PROMIS T-scores are represented on the y-axis.

**Table 1 cancers-16-00547-t001:** Patient characteristics and clinical-, pathological- and treatment-related parameters.

Characteristics	N = 97
Median age y	
IQR	64 (50–72)
Sex	
Female	49 (50%)
Male	48 (50%)
Histological subtype	
Myxofibrosarcoma	26 (27%)
(myxoid) liposarcoma	22 (23%)
MFH/UPS and NOS	21 (22%)
Dedifferentiated/pleomorphic liposarcoma	6 (6%)
Others	6 (6%)
Leiomyosarcoma	6 (6%)
MPNST	4 (4%)
Synovial sarcoma	3 (3%)
Spindle cell sarcoma	3 (3%)
Tumor size cm	
IQR	9 (5–12)
Tumor depth	
Superficial	32 (33%)
Deep	65 (67%)
Grade	
2	55 (57%)
3	42 (43%)
Location	
Upper extremity	19 (20%)
Lower extremity	78 (80%)
ASA score	
0	80 (83%)
1	13 (13%)
≥2	4 (4%)
Surgical margin	
R0	55 (57%)
R1	26 (26%)
R2	-
Amputation	1 (1%)
No surgery (metastasis/dead)	7 (7%)
Unknown *	8 (9%)
Radiotherapy	
Pre-	85 (88%)
Post-	2 (2%)
noRT	7 (7%)
No radiotherapy (metastasis/dead)	3 (3%)
Surgical complications	
Wound infection	10 (10%)
Nerve impairment	2 (2%)
Other	1 (1%)
Disease recurrence	
LR	-
DM	11 (11%)

MFH/UPS; malignant fibrous histiocytoma/undifferentiated pleomorphic sarcoma, NOS; (pleomorphic) soft-tissue sarcomas not otherwise specified, MPNST; malignant peripheral nerve sheath tumor. LR; local recurrence, DM; distant metastasis. * These patients had not been operated on yet, and therefore margins are unknown.

**Table 2 cancers-16-00547-t002:** EQ-5D-5L, PROMIS GH and Physical function scores in eSTS patients.

	T1 (Baseline) *n* = 97	T2 (3 Months) *n* = 81	T3 (6 Months) *n* = 66	T4 (12 Months) *n* = 39
PROMIS PF	46.8 + 1.1	41.2 + 1.1	45.5 + 0.9	49.9 + 0.9
PROMIS GH - Mental - Physical	47.3 + 0.9 46.2 + 1.0	45.8 + 0.9 43.1 + 0.8	47.6 + 0.9 46.5 + 0.9	47.5 + 0.9 50.1 + 0.8
EQ-5D-5L	0.76 + 0.02	0.68 + 0.03	0.81 + 0.02	0.84 + 0.03
EQ-VAS	72.6 + 2.0	70.1 + 2.1	77.6 + 1.7	81.5 + 2.4

Displayed are the mean score and standard error (SE) at baseline (T1), and 3 (T2), 6 (T3) and 12 months (T4) after treatment decision.

**Table 3 cancers-16-00547-t003:** EQ-5D-5L, PROMIS GH and Physical Function scores over time stratified by age.

	T1 (Baseline)		T2 (3 Months)		T3 (6 Months)		T4 (12 Months)	
	<65 Years *n* = 49	≥65 Years *n* = 48	<65 Years *n* = 42	≥65 Years *n* = 39	<65 Years *n* = 35	≥65 Years *n* = 31	<65 Years *n* = 19	≥65 Years *n* = 20
PROMIS PF	47.6 +1.6	45.9 +1.5	41.4 +1.6	40.9 +1.4	46.1 +1.2	44.8 +1.2	53.0 +1.1	47.4 +1.3
PROMIS GH - Mental - Physical	48.1 + 1.4 46.9 + 1.3	46.5 + 1.1 45.4 + 1.4	46.5 + 1.3 43.6 + 1.2	45.1 + 1.1 42.5 + 1.1	49.0 + 1.4 47.1 + 1.2	46.1 + 0.9 45.8 + 1.2	47.9 + 1.5 50.7 + 1.1	47.2 + 1.1 49.4 + 1.3
EQ-5D-5L	0.76 + 0.04	0.75 + 0.04	0.66 + 0.04	0.71 + 0.04	0.78 + 0.03	0.84 + 0.03	0.83 + 0.05	0.82 + 0.03
VAS	70.5 + 3.1	74.8. + 2.5	67.4 + 3.1	73.2 + 2.5	78.2 + 2.5	77.0 + 2.2	82.8 + 2.2	80.2 + 4.2

Displayed are the mean score and standard error (SE) at baseline (T1), and 3 (T2), 6 (T3) and 12 months (T4) after treatment decision.

## Data Availability

The data that support the findings of this study are available from the corresponding author, upon reasonable request.

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
