# Peer review of "Changes in Health-Related Quality of Life following Surgery in Patients with High-Grade Extremity Soft-Tissue Sarcoma: A Prospective Longitudinal Study"

_cancers, 2024, doi:10.3390/cancers16030547_

Round 1

Reviewer 1 Report

Comments and Suggestions for Authors

Very interesting paper that investigates neglected aspects of sarcoma care.

A comment about the choice to anlayze PROMIS PF and GH considering in the same group deep and superficial STS in which impact of surgical resection and indication to RT is different. Furthermore if we  consider that in the elderly are more common MFS with an high rate of presentation as superficial tumours. I think it would be more adequate to split in subgroups to avoid bias in data analysis and interpretation.

A question about the study group. In Table 1 is reported a 33% of superficial STS, but 90% of patients received RT.  So the indication for RT has been given for almost the entire cohort, regardless the location.  Can you explain this indication? Is it due to the size or is it a standard approach?

Author Response

  1. A comment about the choice to analyze PROMIS PF and GH considering in the same group deep and superficial STS in which impact of surgical resection and indication to RT is different. Furthermore if we  consider that in the elderly are more common MFS with an high rate of presentation as superficial tumours. I think it would be more adequate to split in subgroups to avoid bias in data analysis and interpretation.

Response: We agree with the reviewer that there might be potential variation in HRQoL of patients with deep-seated vs superficial tumors considering the indication of different treatment modalities. However, the differentiation in treatment was not shown in our population, where 88% (n=85) of patients received pre-operative radiotherapy. Nonetheless, we explored whether we could perform a reliable analysis to compare HRQoL over time of patients with deep-seated vs superficial tumors. It appeared that only HRQoL data of 9 of the 32 patients with superficial tumors on t=4 (one year after diagnoses) were available, and of 31 of the 65 patients with deep-seated tumors. Because the subgroups of patients with superficial and deep seeded tumors are too small at T4, it is impossible to make a reliable analysis and a valid interpretation of the differences in HRQoL between both groups of patients over time. Therefore, we decided not to include an analyses in which we split up the HRQoL data for superficial and deep-seated tumors in our revised manuscript. We added this limitations in the manuscript with track changes on page 10, lines 308-309.

  1. A question about the study group. In Table 1 is reported a 33% of superficial STS, but 90% of patients received RT.  So the indication for RT has been given for almost the entire cohort, regardless the location.  Can you explain this indication? Is it due to the size or is it a standard approach?

Response: Thank you for observing this information. The indication for additional radiotherapy in the participating centers (and, as a matter of fact, according to international guidelines) is based upon an “and/or” evaluation. Perioperative RT is offered to sarcomas of grades II/II (100% of our population) and/or deep seated (67%) and/or large than 5cm (again 100%). The “and/or” option explains your observation.

Reviewer 2 Report

Comments and Suggestions for Authors

The authors made a prospective study to sequentially evaluate the quality of life of individuals who underwent treatment for extremities soft tissue sarcomas. 

The title is more general than the study is. The authors have to specify SOFT TISSUE Sarcoma in the title.

Please also correct the tipo error "Ppatients" in the title.

The authors use the VALUE-PERSARC trial, which is a wonderful project; however, using that trial brought many biases to this study. The authors should include more data to correct those biases and obtain more reliable results. Some critical surgical information needs to be included. 1)how many patients were amputated? It must be informed if no one was amputated or if it was an exclusion criterion. Were only patients with preserved limbs included? 2) Patients who presented local relapse during the first year must be identified, and a separate analysis is essential since a local relapse can influence the quality of life. 3) there were surgical complications? If yes, those complications can influence the results. 4) Lower limb procedures have distinct post-operative evolution from the upper limbs. Comparing the quality of life of upper limbs tumor resection and lower limbs is a bias. 

LINE 294: The authors recognize the use of the date of treatment decisions as a bias to count the time instead of the date of tumor resection. "Additionally, the specific timepoints  to assess HRQoL were not linked to the date of surgery but to the treatment consultation, potentially causing differences in the HRQoL scores (at 3, 6 and 12 months) due difference in care trajectories of patients who did not undergo RT or underwent post-operative RT."  There is much variability during post-operative procedures. The quality of life is very different after one month and four months. If the tumor resections occurred in the same time interval after the treatment consultation, it reduces the bias; in this case, the authors must inform how long after the treatment consultation the surgery was performed.

The authors included metastatic patients; don't you believe it brings a strong bias to the study?

The figures' descriptions were out of logical order; with three figures and four figures' descriptions, one must be included.

I am sorry, but I had difficulty understanding what the term "clinically relevant differences" means. Is this related to p values? Because I did not see any p-values compared to the general population. Could you explain?

Round 2

Reviewer 2 Report

Comments and Suggestions for Authors

The authors presented answers to the questions.